# Geographic barriers to care persist at the community healthcare level: Evidence from rural Madagascar

**Michelle V. Evans**[1]*, **Tanjona Andréambeloson**[2], **Mauricianot Randriamihaja**[2], **Felana Ihantamalala**[2,3], **Laura Cordier**[2], **Giovanna Cowley**[2], **Karen Finnegan**[2], **Feno Hanitriniaina**[2], **Ann C. Miller**[3], **Lanto Marovavy Ralantomalala**[2], **Andry Randriamahasoa**[2], **Bénédicte Razafinjato**[2], **Emeline Razanahanitriniaina**[2], **Rado J. L. Rakotonanahary**[2], **Isaïe Jules Andriamiandra**[4], **Matthew H. Bonds**[2,3], **Andres Garchitorena**[1,2]

**1** MIVEGEC, Univ. Montpellier, CNRS, IRD, Montpellier, France, **2** NGO PIVOT, Ranomafana, Ifanadiana, Madagascar, **3** Department of Global Health and Social Medicine, Blavatnik Institute at Harvard Medical School, Boston, MA, United Sates of America, **4** Direction de Soins de Santé de Base, Madagascar Ministry of Public Health, Antananarivo, Madagascar

☯ These authors contributed equally to this work.
* mv.evans.phd@gmail.com

**Data Availability Statement:** All code and data needed to reproduce the analysis are available in a figshare repository (https://doi.org/10.6084/m9.

## Abstract

Geographic distance is a critical barrier to healthcare access, particularly for rural communities with poor transportation infrastructure who rely on non-motorized transportation. There is broad consensus on the importance of community health workers (CHWs) to reduce the effects of geographic isolation on healthcare access. Due to a lack of fine-scale spatial data and individual patient records, little is known about the precise effects of CHWs on removing geographic barriers at this level of the healthcare system. Relying on a high-quality, crowd-sourced dataset that includes all paths and buildings in the area, we explored the impact of geographic distance from CHWs on the use of CHW services for children under 5 years in the rural district of Ifanadiana, southeastern Madagascar from 2018–2021. We then used this analysis to determine key features of an optimal geographic design of the CHW system, specifically optimizing a single CHW location or installing additional CHW sites. We found that consultation rates by CHWs decreased with increasing distance patients travel to the CHW by approximately 28.1% per km. The optimization exercise revealed that the majority of CHW sites (50/80) were already in an optimal location or shared an optimal location with a primary health clinic. Relocating the remaining CHW sites based on a geographic optimum was predicted to increase consultation rates by only 7.4%. On the other hand, adding a second CHW site was predicted to increase consultation rates by 31.5%, with a larger effect in more geographically dispersed catchments. Geographic distance remains a barrier at the level of the CHW, but optimizing CHW site location based on geography alone will not result in large gains in consultation rates. Rather, alternative strategies, such as the creation of additional CHW sites or the implementation of proactive care, should be considered.

figshare.20412624.v1). In addition, geographic routing data are available to visualize and download on Pivot's research dashboard (https://research.pivot-dashboard.org/).

**Funding:** This work was supported by internal funding from PIVOT (https://www.pivotworks.org/), which provided salary for TR, MR, FI, LC, GC, KF, FH, LMR, AR, BR, ER, RJLR, and MHB. It was also supported by a grant from the Agence Nationale de la Recherche (Project ANR-19-CE36-0001-01), granted to AG, which supported AG, MR, and MVE. The funders had no role in study design, data collection and analysis, decision to publish, or preparation of the manuscript.

**Competing interests:** I have read the journal's policy and the authors of this manuscript have the following competing interests: TR, MR, FI, LC, GC, KF, FH, LMR, AR, BR, ER, RJLR, and MHB receive salary from PIVOT, currently or in the past. IJA is employed by the Madagascar Ministry of Public Health. These authors confirm that their competing interests will not alter adherence to PLOS Global Public Health policies on sharing data and materials.

# Introduction

In 2012, the UN General Assembly declared universal health coverage key to sustainable human and economic development. However, the majority of the world's population still lacks access to essential heath services, including primary care [1]. There are multiple barriers to primary care access, including financial costs, low provider to population ratios, and geographic distance [2, 3]. Geographic distance to primary health clinics (PHCs) is especially problematic in contexts where the transportation network is sparse and the primary mode of transportation is non-motorized [4]. In these primarily rural regions, the use of primary care decreases exponentially for populations living at increasing distances from PHCs, known as the "distance-decay" effect [5–9]. Low access in turn negatively affects population health metrics such as maternal mortality [10] or hospitalization rates for severe malaria [11].

Community health programs, often implemented via community health workers (CHWs), have emerged as a potential solution to increase healthcare access in geographically isolated communities [12]. Usually residents of these communities, CHWs are located closer to their patients than PHCs and provide basic maternal and child care, as well as a range of services such as HIV and TB treatment. CHW programs became a focus of international development funding organizations following the Alma Alta declaration in 1978 and were implemented in countries around the world. CHWs have since played an integral role in national health systems and several universal health coverage strategies [13–15]. Although located in closer proximity to underserved communities than other levels of the health system (e.g. PHC, district hospital), CHWs can be responsible for large geographic catchments or isolated, rural populations, limiting their ability to adequately serve everyone. As such, the World Health Organization recommends that community health programs adapt CHW catchments according to the local geography, including the proximity of households and population density [16]. Yet, it is unknown whether a meaningful 'distance-decay' relationship—similar to that observed for health facilities—can persist at the community health level, where distances to seek care are significantly shorter. Whether this relationship exists has significant implications for how CHW systems are designed geographically, and if and how community health programs should be optimized to the local geography.

Measuring the impact of geographic barriers on community health poses particular challenges. Studies on barriers to healthcare typically use data from national surveys, such as Demographic Health Surveys or Multiple Indicators Cluster Surveys (e.g. [17, 18]). Household coordinates in these surveys are randomly displaced to protect participants' anonymity [19], preventing the assessment of geographic barriers at small spatial scales [20]. Dedicated surveys have measured households' reported time to different health providers and the impact on healthcare access [21, 22], but these rarely assess the impact on CHW access (but see Ahmed et al. [23] for a review of the few, existing studies) and could be subject to biases in people's perception of time. An alternative is the use of geographic information from routine health system data. While PHCs typically keep records of a patient's residence to the lowest administrative level [9], this information is not systematically recorded in CHW registries. Studying the impact of geographic barriers at the community level therefore requires new methods and data if we are to optimize community health programs for the local geography of CHW catchments.

Madagascar is illustrative of the challenges to and opportunities for achieving universal health coverage by improving the design of community health systems. Following a history of French colonial occupation and political instability post-independence, Madagascar has witnessed little economic growth over the past half-century, severely limiting investment in public health, with negative consequences for healthcare access [24, 25]. There are fewer than 2

medical doctors and fewer than 3 nurses or midwives per 10,000 people in the country [26], and 43% of the population lives more than an hour walking time from a PHC, a common threshold for defining adequate access [4]. In contrast, the WHO recommended index for achieving universal health coverage is 44.5 doctors, nurses, or midwives per 10,000 people [27]. Nationally, health professionals cite common barriers to healthcare access seen in other settings such as the cost of medicines, geographic distance, mistrust of the health system, and unsanitary conditions at health facilities [28]. In addition, studies of healthcare access in rural districts of Madagascar have identified geographic distance to PHCs as a key barrier to primary care, even when the health system is strengthened and financial barriers have been reduced [29, 30]. Consequently, healthcare seeking rates for children under 5 years old are low, ranging from 40–60% nationally [31], and there is an opportunity for a strengthened community health program to help meet these needs. The national community health policy requires that two volunteer CHWs be locally selected and responsible for each fokontany, the smallest administrative area in Madagascar (mean population of approx. 1200) [32]. However, fokontany range in size from 1.5 km$^2$ to 3747 km$^2$. Given this range of catchment sizes, populations in rural Madagascar may continue to face geographic barriers when seeking care at the CHW level, which could limit the Malagasy health system's ability to provide adequate coverage.

Here, we examined the impact of geographic distance to CHWs on the use of CHW services for children under 5 years in the rural district of Ifanadiana, southeastern Madagascar. Ifanadiana district has been established as a model district for universal health coverage by the government of Madagascar, implemented in partnership with the healthcare non-governmental organization (NGO) Pivot (see the 'Study area and intervention' section), with a particular focus on strengthening community healthcare. We build on previous work where we obtained accurate estimates of distance to PHCs and CHWs for every household in the district using a high-resolution spatial dataset that includes all households and transportation networks used by residents in the district [33]. From this dataset, we calculated an aggregate measure of geographic dispersion from the CHW site for each fokontany in our study area, and we assessed its relationship with all-cause child consultations with CHWs, controlling for relevant factors. We then explored two potential solutions to geographic barriers for isolated communities in Ifanadiana: optimizing a single CHW site location with regards to distance and installing an additional second site per catchment.

## Methods

### Study area and intervention

Madagascar is an island nation of approximately 587,041 km2 with a population of 28.1 million people located off the eastern coast of the African continent. Sixty-one percent of the population lives in rural areas [34], a mosaic of agriculture lands, forest, and savanna. The road network is sparse with only 10% characterized as passable year-round [35], and non-motorized transport by foot or bike is the norm. The majority of the population are agricultural producers, growing primarily rice, and 81% of the population lives below the international poverty line [34]. National government expenditure on healthcare is 19.75 USD, less than half the expenditure of neighboring countries [34], and many indicators of child health are below Sustainable Development Goal 2030 targets (Table 1).

Following national policy, each commune (subdivision of a district with ∼15 000 people) in Madagascar has at least one primary health center (PHC) that provides primary care for people of all ages. Each fokontany (subdivision of a commune with ∼1200 people) has two community appointed CHWs that provide integrated community case management (iCCM) [32]. Recent national policy recommends one CHW per 50 households in rural settings, but

**Table 1. Comparison of key child health metrics for Ifanadiana district and Madagascar.**

|  | Ifanadiana[a] | Madagascar[b] |
| --- | --- | --- |
| Immunization coverage (12–23 months) | 44% | 41% |
| Infant mortality rates (per 1000) | 70 | 40 |
| Child stunting (less than 5 years) | 45% | 50% |
| Rates of childbirth at health facilities | 31% | 39% |
| Rates of care seeking for fever (less than 5 years) | 51% | 48% |

[a]IHOPE Longitudinal Survey 2018,

[b] Multiple Indicator Cluster Survey 2018

the policy has not yet been implemented [36]. Community health care, including diagnoses and distribution of limited medicines, takes place at the community health site of each fokontany, a static location, although more recent policies (2023 onward) recommend home visits. iCCM focuses primarily on malaria, diarrheal disease, and respiratory infections [36]. Those with severe symptoms or other diseases are referred to PHCs. CHWs are unpaid volunteers, making only marginal profits from the sale of certain medicines, and receive monthly supervision at PHCs.

Ifanadiana is a rural district located in the Vatovavy region in southeastern Madagascar. It has a population of approximately 200,000 people, distributed in 13 communes and 195 fokontany. Fokontany in the district range in size from 1.94–138.95 km$^2$, with an average size of 20.36 km$^2$ (S1 Table). The district is topographically complex, with areas of steep terrain, and landcover is a mosaic of rice fields and savanna. The transportation network consists of one paved road (<1% of the total road network in the district) that travels east-west and a secondary, unpaved road that follow a north-south axis [33]. The rest of the transportation network is tertiary roads or footpaths. Most of the population practices agricultural and nearly three-quarters live in extreme poverty [37].

In 2014, the Madagascar Ministry of Public Health (MMoPH) and the NGO Pivot began a partnership to strengthen the community health system as part of a broader intervention at all levels of care to achieve universal health coverage in Ifanadiana District (S2 Table). Public health facilities provide over 80% of healthcare in the district, primarily at PHCs, but utilization rates decrease by 50% for every increase in travel time of 1 hour to the PHC [9]. The strengthened community health program aimed to address this and revolved around an enhanced standard of care model which included providing CHWs with additional training; frequent on-site supervision by a team of nurses; the construction of a physical building for consultations in partnership with local communities; support to the supply chain, with medicines free of charge; and a modest stipend [32]. As of February 2022, this intervention had been implemented in seven communes, equivalent to about half the district, with plans to expand to the entire district by 2023.

## Data collection

**Distance to primary health centers and community healthcare workers.** For each fokontany, we estimated the average distance of all households in the fokontany to two types of health facilities, primary health centers (PHCs) and community health sites where CHWs are based. For this, we used a high-resolution spatial dataset available on OpenStreetMap, which was collected via participatory mapping (described in detail in Ihantamalala et al. [33]). This dataset included all paths, roads, buildings, and residential areas (defined as groups of four or more buildings) within Ifanadiana District. Overall, the dataset included over 100,000

buildings, 20,000 km of footpaths and more than 5000 residential areas. For each residential area, we estimated the network distance to the nearest health facility of each type using the Open Source Routing Machine algorithm (http://project-osrm.org/), along our mapped transport network of roads and paths. We then aggregated these values to the fokontany level by taking a weighted mean for all residential areas within a fokontany (residential areas were weighted by the number of buildings they comprised), resulting in an average household distance to the nearest PHC and community sites. We refer here to the average household distance to the fokontany's community health site as geographic dispersion.

**Survey data.** Household wealth is strongly associated with higher access to healthcare in Ifanadiana [38]. Therefore, we calculated the average wealth score for each fokontany using data from the IHOPE cohort, which includes longitudinal surveys at 80 clusters across Ifanadiana district, conducted every two years from 2014–2018 [37]. The sampling scheme was a two-stage cluster sampling design, where 40 clusters were sampled at random within each of two strata, within the Pivot initial catchment and the rest of the district. Twenty households per cluster were then randomly selected to be surveyed. The IHOPE cohort was implemented by the Madagascar National Institute of Statistics and was based primarily on the internationally validated Demographic and Health Surveys (DHS). Further details on the longitudinal survey can be found in Miller et al. [37] and Ezran et al. [39]. A wealth score was calculated for each of the 1600 households through a principal components analysis of household assets following standard DHS methods [40]. Because household wealth scores in Ifanadiana are very stable over time (S1 Fig), we summarized this value as the mean for each cluster across survey years. We then used inverse-distance weighting to interpolate these values from 80 clusters across the study area [41], using leave-one-out cross-validation to optimize the power used to weight the distance, via the spatstat package [42]. To aggregate this value to the scale of the fokontany, we extracted the wealth score at the coordinates of the four primary residential areas of each fokontany and assigned the mean of these values to the fokontany.

**Community health system data.** To study the impact of geographic barriers on CHW utilization, we obtained the monthly number of CHW consultations for children under 5 years old from the CHW monthly reports to the District Office. All CHWs in these fokontany receive additional training and supervision from Pivot, including training on data reporting methods. In addition, mobile teams provide regular supervision and trained staff evaluate subsets of data for accuracy. These CHW reports, therefore, are a unique source of high quality data at the community level. We included data from 80 fokontany in seven communes in Ifanadiana, from May 2018—May 2021, except for one commune (Ambohimanga du Sud), where CHW support began in June 2020. Consultations here refer to all-cause consultations for children under 5 years old who are residents of that fokontany. Per national policy, CHW perform iCCM, and therefore only treat children under 5 years old who present with malaria, diarrheal disease, or respiratory infection. To obtain per capita utilization rates (per child under 5), we used population data collected by Pivot in 2021 as part of the proactive CHW program. We interpolated population estimates to other years assuming a 2.8% growth rate, following MMoPH methods.

In addition, we collected information to control for characteristics of the CHWs that may influence healthcare seeking behavior. This included the average age of the CHWs (median: 43.5, range: 20–63) and the gender of the CHWs in a community site (all men, all women, mixed). This data is routinely collected by the Pivot Community Health team. Because CHW composition was stable over the time period of this study (May 2018—May 2021), we assumed that these CHW characteristics were constant for the duration of our study. We also accounted for the introduction of a pilot initiative of proactive care in one of the communes (Ranomafana) in October 2019. Pivot teams also conduct monthly on-site supervisions of CHWs,

observing multiple consultations and recording the percentage of cases of malaria, pneumonia, and diarrhea correctly managed (diagnosis, treatment, recommendations, etc.). From these measurements, we calculated the proportion of consults correctly treated across all three diseases as a metric of quality of care. This variable was used in a supplemental analysis to examine the impact of whether the CHW quality of care modified the relationship between our geographic proxies and healthcare access.

## Statistical analysis of geographic barriers to community healthcare

We tested for an effect of a fokontany's geographic dispersion from the CHW site on monthly CHW consultation rates using a generalized linear mixed model. To control for factors that are known to influence CHW use rates, we included the fokontany's average household distance to the nearest PHC, the fokontany's wealth score, and characteristics of the CHWs working in the community site (age and sex). All continuous predictor variables were scaled and centered to facilitate model convergence. We standardized counts of consultations per month per fokontany into the number of monthly consultations per 1000 children under 5 years old. The model was fit with a zero-inflated negative binomial distribution with a log-link, including the fokontany and month of sampling as crossed random effects to account for the repeated sampling design of the data [43]. We assessed the fitted model for variable collinearity, residual uniformity, and overdispersion. We also conducted a supplemental analysis on the subset of the data for which we had information on the quality of care (80.2%). The statistical analysis was identical to our primary analysis except for the inclusion of the additional quality of care variable. The analyses were conducted using the glmmTMB [44] and DHARMa [45] packages in R v. 4.1.1 [46].

## Potential solutions to geographic barriers

Following the statistical analysis estimating the relationship between geographic dispersion and consultation rates, we considered two solutions to geographic barriers to CHW access: optimizing the location of a single community health site and installing a second community health site per fokontany in an optimized location. We assessed whether relocating the community health site or optimizing the location of a second site to minimize geographic dispersion could substantially improve CHW consultation rates in the study area. To optimize the location of one CHW site, we identified the location within an existing residential area in that fokontany that resulted in the lowest geographic dispersion value. The optimal location was identified by calculating the geographic dispersion value for the fokontany from each residential area, following the method above, and selecting the residential area with the lowest value. To optimize the location of a second site, we calculated the geographic dispersion value for each fokontany with one site in the current location and the second site within an existing residential area. We assumed each household would use the community site closest to their residence within their fokontany, and calculated the average household distance to the closest community site for each potential new second location. The optimal second location was the residential area that resulted in the lowest average geographic dispersion value for the fokontany. We then used the fit statistical model to predict the change in consultation rates given new optimized site locations and reduced estimates of geographic dispersion, and compared predicted consultation rates at optimized locations to predicted rates at present locations.

We created an online dashboard to share the results of these optimization exercises for the full district of Ifanadiana (https://research.pivot-dashboard.org). This interactive cartographic tool allows public health stakeholders to identify current, optimal single, and optimal second CHW site locations for all 195 fokontany in Ifanadiana district. In addition, they can explore

the predicted change in CHW consultation rates for both potential solutions for the 80 fokontany included in the statistical analysis. To ensure adoption of this tool by CHW supervisors at Pivot, we conducted a training on tool use following its establishment. The tool is available in English and French, the latter of which is spoken by CHW supervisors.

### Ethical statement

Use of aggregate monthly consultation counts from MMoPH community health data for this study was authorized by the Medical Inspector of Ifanadiana district. It was deemed non-human subjects research by Harvard University's Institutional Review Board. The IHOPE longitudinal survey implemented informed consent procedures approved by the Madagascar National Ethics Committee and the Madagascar Institute of Statistics, specifically including written consent from participants aged 15 years or over. Household-level de-identified data from the IHOPE survey were provided to the authors for the current study.

This research was conducted by an international team consisting of researchers, health practitioners, and international development workers with differing relations with and identities relative to the region and community of study. We explore how these power dynamics influenced our research process in a reflexivity statement in the supplement following Morton et al. [47] (S1 File).

## Results

### Geographic barriers to community healthcare

CHWs conducted 73,857 consults from May 2018—May 2021, with an average annual consultation rate of 1.73 (1.51 *sd*) consultations per capita. These consultations are all-cause consultations for children under-5 seen by a CHW. Data was missing for 62 out of 2436 month-fokontany combinations, with one fokontany missing ten months of data, but the majority of fokontany (56) missing none. Geographic dispersion estimates ranged from 0.21 to 4.12 km, with a mean of 2.06 km (Fig 1). Geographic dispersion was a function of both the spatial pattern of residential zones relative to the community health site and the overall absolute size of the fokontany (Fig 1B and 1C). The fokontany with the lowest geographic dispersion had 93.8% of the population living within 1 km of the community health site, with a maximum distance of 1.65 km. In contrast, the fokontany with the highest geographic dispersion had 14.8% of the population living within 1 km of the community health site, with a maximum distance of 8.34 km.

Our statistical model suggested that the geographic dispersion of a fokontany is associated with CHW consultation rates. Specifically, consultation rates were lower in fokontany where residential areas are more geographically dispersed from CHW sites (Table 2 and Fig 2A). An increase in geographic dispersion of 1 km corresponded to a decrease in monthly consultation rates of 28.1% (95% CI: 19.0% - 36.2%, Table 2). For example, while keeping all other variables unchanged and at their mean value, an increase in geographic dispersion from 1 km to 4 km corresponded to a predicted decrease in annual consultation rates from 2.18 to 0.81 per capita. We also found a trend for higher CHW consultation rates in fokontany that are further from PHCs (Table 2 and Fig 2B), but this trend was not significant (IRR = 1.019, 95% CI = 0.988–1.052). Wealthier fokontany had lower consultation rates than fokontany with lower socio-economic levels (IRR = 0.686, 95% CI = 0.533–0.882, Table 2). Consultation rates were higher in fokontany that had mixed gender CHWs (IRR = 1.514, 95% CI = 1.154–1.984) or all men teams (IRR = 1.423, 95% CI = 1.06–1.919), compared to teams of all women (Table 2). Our supplemental analysis found that these results were robust to the inclusion of the quality of care provided by CHWs in the model (S2 Fig).

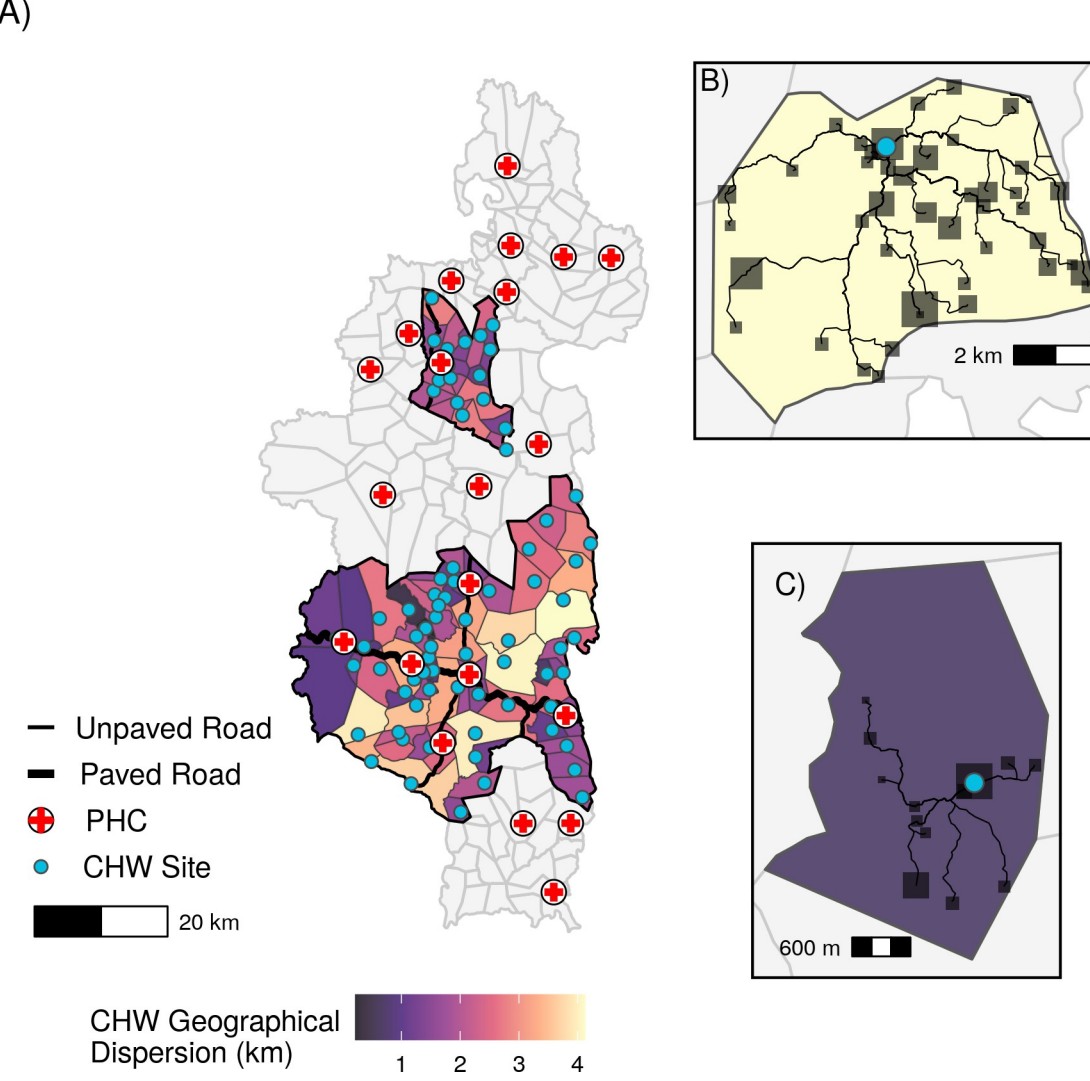

**Fig 1. Spatial patterns of geographic access to health care at community health sites in Ifanadiana.** A) Geographic dispersion of the fokontany population from the CHW (in km), based on the transportation network and location of all residential areas. Panels B) and C) show the routes from residential areas to CHW sites in a fokontany with high geographic dispersion (B) and low geographic dispersion (C). Blue points represent the community health site for that fokontany and shaded squares represent residential areas, with larger squares representing areas with more buildings. CHW: community health worker, PHC: primary health center. Data Sources: MMoPH (administrative boundaries), Pivot (health center locations), OpenStreetMap (transportation routes, buildings). All maps have been created from spatial datasets using the R programming language.

## Impact of single optimized community health site placement on CHW consultations

Given the association between CHW geographic dispersion in a fokontany and CHW consultation rates, we explored whether optimized CHW site placement could be a potential solution to reduce geographic barriers to community health access. Forty-six fokontany (of 80) had an existing community site that was within 500m euclidean distance of the optimal site placement (Fig 3), resulting in an average reduction in geographic dispersion of 0.062 km. For this exercise, we considered these forty-six instances as fokontany where the CHW site was already optimally located. Of the 34 fokontany with an optimal site placement further than 500m from

**Table 2. Results of statistical model estimating effects of geographic barriers on CHW consultation rates.** Incidence rate ratios (IRR) of variables included in the model. Following Wasserstein et al. [48], we have chosen to provide 95% confidence intervals and not to dichotomize these continuous measures into thresholds of significance (e.g. p-values). CHW: community health worker, PHC: primary health center.

| | Mean IRR | 95% CI |
|---|---|---|
| **Geographic Variables** | | |
| CHW Geographic Dispersion (km) | 0.719 | 0.638–0.811 |
| Distance to PHC (km) | 1.019 | 0.988–1.052 |
| **Confounding Variables** | | |
| Wealth Score | 0.686 | 0.533–0.882 |
| CHW Age (year) | 1.000 | 0.991–1.015 |
| CHW Gender- All Men (compared to all women) | 1.428 | 1.062–1.919 |
| CHW Gender—Mixed (compared to all women) | 1.514 | 1.115–1.984 |
| Proactive Care | 1.471 | 1.297–1.668 |
| **Model Structure** | | |
| Zero-inflation Intercept | 0.026 | 0.020–0.033 |

the existing site, four had a PHC located within their borders, and the optimal location was in the same residential area as the PHC. For the remaining 30 fokontany, the optimal CHW site placement was in a different geographic location than the current CHW site or PHC (Fig 3). The quartile of fokontany with the highest geographic dispersion (above 3.3 km) experienced an average reduction in geographic dispersion of 19.1% and absolute change in geographic dispersion of 0.71 km when calculating geographic dispersion using the optimal site placement compared to the current CHW site location. In contrast, the quartile of fokontany with the lowest geographic dispersion (below 0.90 km) experienced an average reduction in geographic dispersion of 15.3% and absolute reduction in geographic dispersion of 0.31 km when calculating geographic dispersion using the optimal site placement compared to the current CHW site location. In some instances, particularly for large fokontany, the distance between optimal and current site locations was several kilometers, but the change in geographic dispersion was small (Fig 3). This was because the size of the fokontany and road network necessarily placed a limit on the lowest geographic dispersion possible when only placing one CHW site.

Using the fitted GLMM, the predicted change in annual consultation rates given optimal CHW site location was a modest increase of 0.13 consultations per year per child, or a 7.4% increase in annual consultation rates, across all 80 fokontany. Focusing on only the 30 fokontany with an optimal site placement more than 500m from the existing site, we predicted a 15.8% increase in consultation rates with optimized CHW site placement. For the quartile of fokontany that had the highest geographic dispersion, optimal CHW site placement resulted in a 15.2% increase in consultation rates compared with current CHW site placement (Fig 4). For the quartile of fokontany with the lowest geographic dispersion, optimal CHW site placement resulted in a 12.6% increase in consultation rates compared with current CHW site placement (Fig 4).

## Impact of a second community health site on CHW consultations

The second solution we explored was the addition of a second community site to each fokontany in a location optimized to reduce the geographic dispersion of that fokontany. The optimal location of the second community site in four fokontany was the same location as a PHC

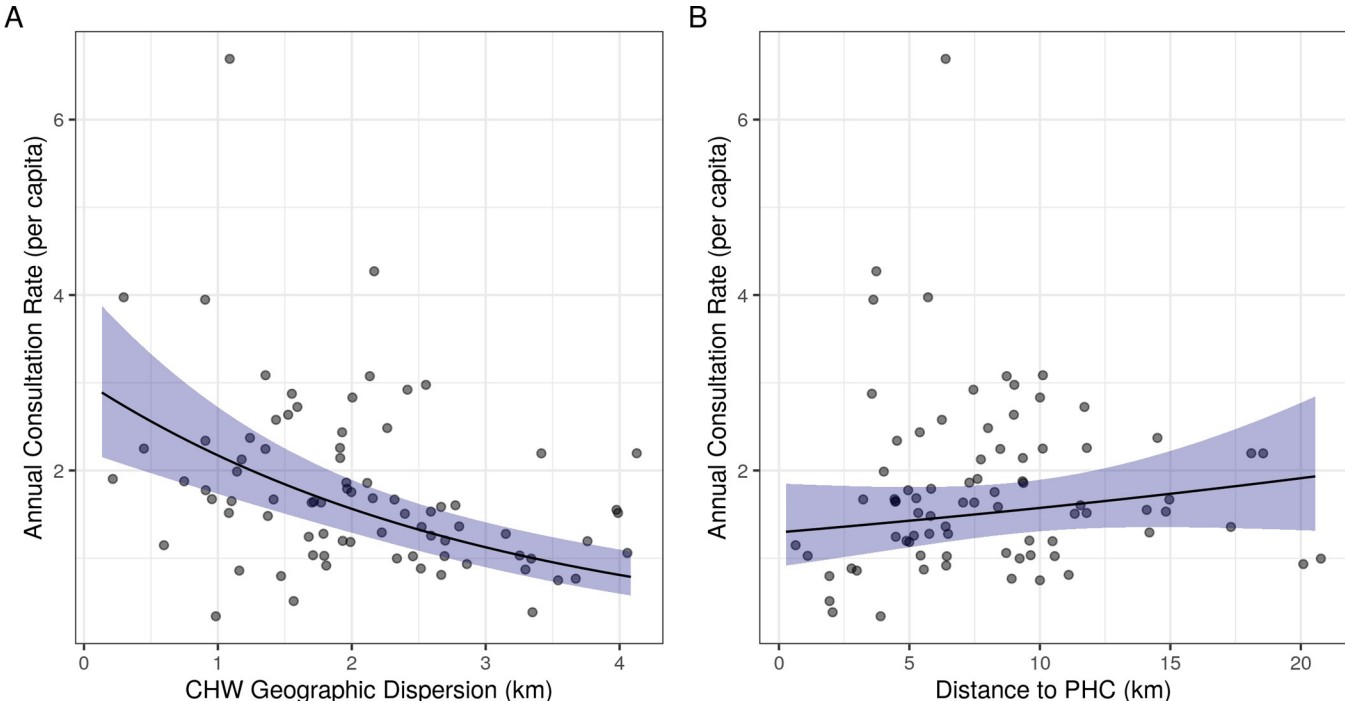

**Fig 2. CHW consultation rates for children under 5 years decrease with increasing CHW geographic dispersion and increase with increasing distance to the nearest PHC.** Lines represent the estimated marginal mean effects of each variable with 95% confidence intervals from the fitted model, holding all other variables constant at their mean value. Points represent the mean annual CHW consultation rate for each fokontany from May 2018—May 2021. CHW: community health worker, PHC: primary health center.

(Fig 5), and so we did not include them in further analyses. The remaining 76 fokontany had a second site on average 2.29 km (0.98 km *sd)* from the existing site. On average, 42.2% of the population of each fokontany lived closer to the second site than the current site, representing the potential population served by a second community health site. The quartile of sites with highest geographic dispersion experienced an average reduction in geographic dispersion of 38.2%, or 1.27 km, following the addition of a second site (Fig 5). The quartile of sites with the lowest geographic dispersion experienced and average reduction in geographic dispersion of 48.8%, or 0.44 km, following the addition of a second site (Fig 5).

After applying the fitted GLMM, we found that the addition of a second site was predicted to increase the average annual consultation rate from 1.81 consults per capita to 2.38 consultations per capita, an increase of over 30%, for all 76 fokontany. Considering the quartile of fokontany that had the highest geographic dispersion, the creation of a second site resulted in a predicted 49.1% mean increase in consultation rates (Fig 4). For the quartile of fokontany with the lowest geographic dispersion, the creation of a second site resulted in a predicted 27.3% mean increase in consultation rates (Fig 4).

## Discussion

Geographic distance to health facilities is one of the main barriers to healthcare access, particularly in rural communities that often live several hours travel time from the nearest clinic. Community health workers (CHWs) have worked for decades within these isolated communities to provide basic health services, especially to children under five years old, and to increase geographic access to primary care. However, even at this finer spatial scale, distance to CHWs may continue to negatively impact healthcare access. A renewed push for community health

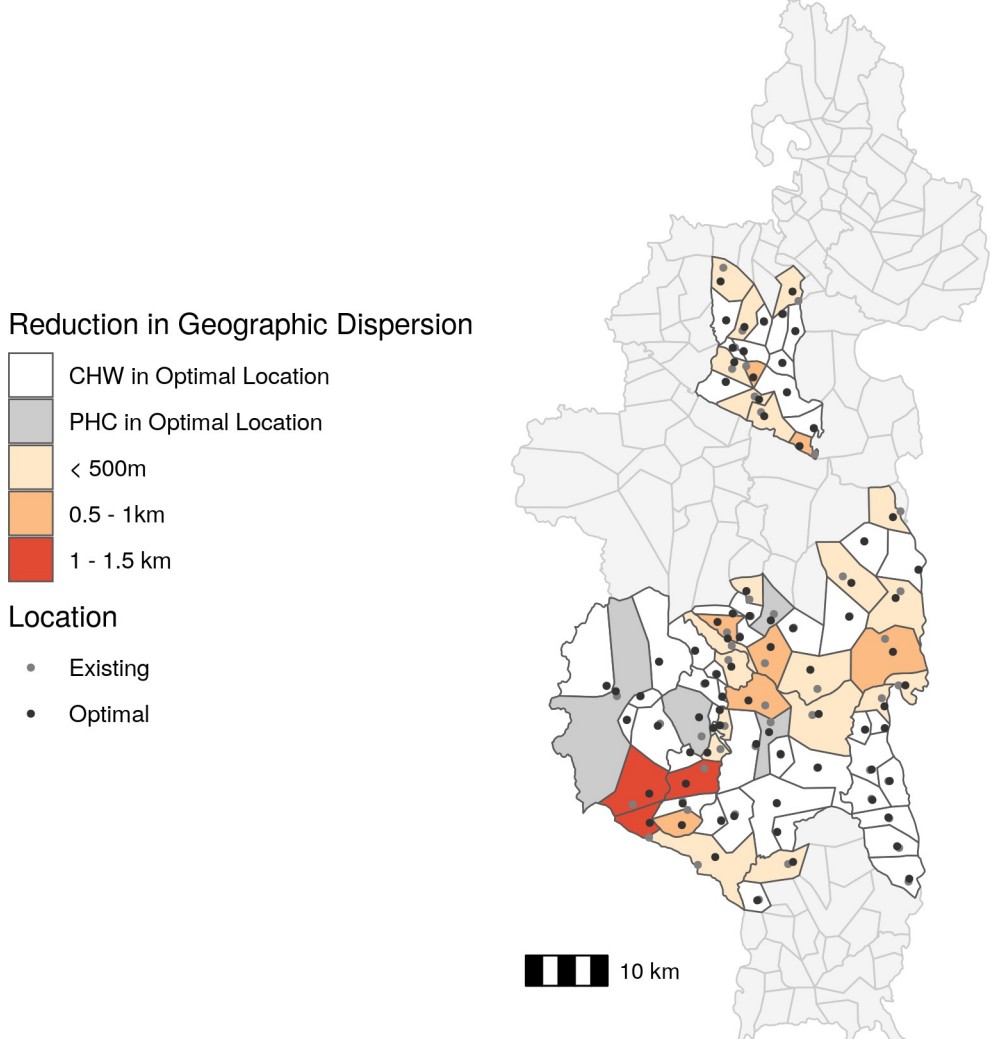

**Fig 3. The majority of existing CHW sites are located near the optimal location.** Absolute change in CHW geographic dispersion based on optimal CHW site location compared to existing CHW site location. The fill color represents the reduction in geographic dispersion and points represent existing site locations (gray) and optimal site locations (black). CHW: community health worker, PHC: primary health center. Data Sources: MMoPH (administrative boundaries), Pivot (health center locations), OpenStreetMap (transportation routes, buildings). All maps have been created from spatial datasets using the R programming language.

has translated into updated international guidelines to optimize community health programs in order to maximize their impact, including recommendations to adapt programs to the local geography of CHW catchments. Using a comprehensive transport network dataset, we examined whether geographic barriers persist at the CHW level for child consultation rates in a rural district of Madagascar with a strengthened community health program. We found that consultation rates were lower in fokontany where households lived further from the CHW site (i.e. higher geographic dispersion), even after controlling for factors such as wealth and healthcare provider characteristics. Optimizing the location of these CHW sites to minimize geographic dispersion alone had minimal impact on predicted consultation rates. However, the addition of a second community site was predicted to increase consultation rates by nearly

## Optimized Single CHW Location

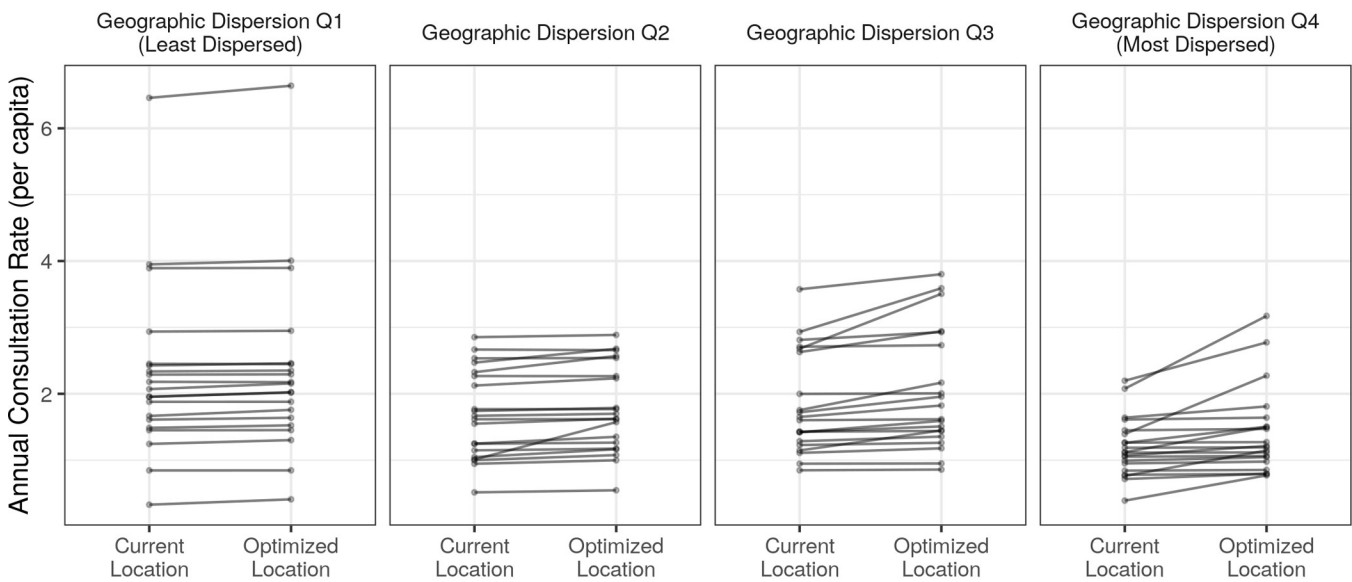

## Installing Second CHW Location

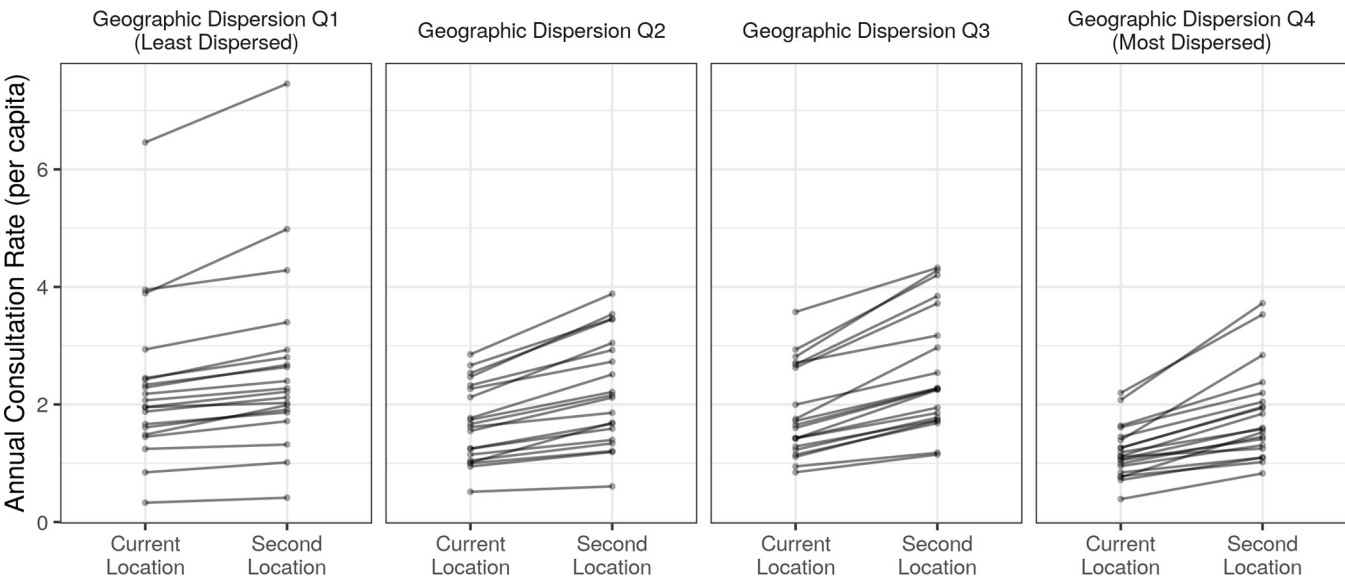

**Fig 4. Both proposed solutions have the strongest predicted improvement in consultation rates in the most geographically dispersed fokontany.** The predicted change in annual CHW consultation rates following the optimization of a single CHW site (top row) or the addition of a second CHW site in an optimized location (bottom row). Each line represents the predicted change for one fokontany and fokontany are grouped into quartiles determined by their geographic dispersion estimates (Q1: least dispersed, Q4: most dispersed).

50% in the most geographically dispersed fokontany. While geographic barriers clearly play a role in limiting healthcare access in this setting, our results suggest that optimizing the location of a single community site will not reduce geographic barriers. Rather, alternative solutions, such as installing additional sites in more dispersed fokontany or the implementation of

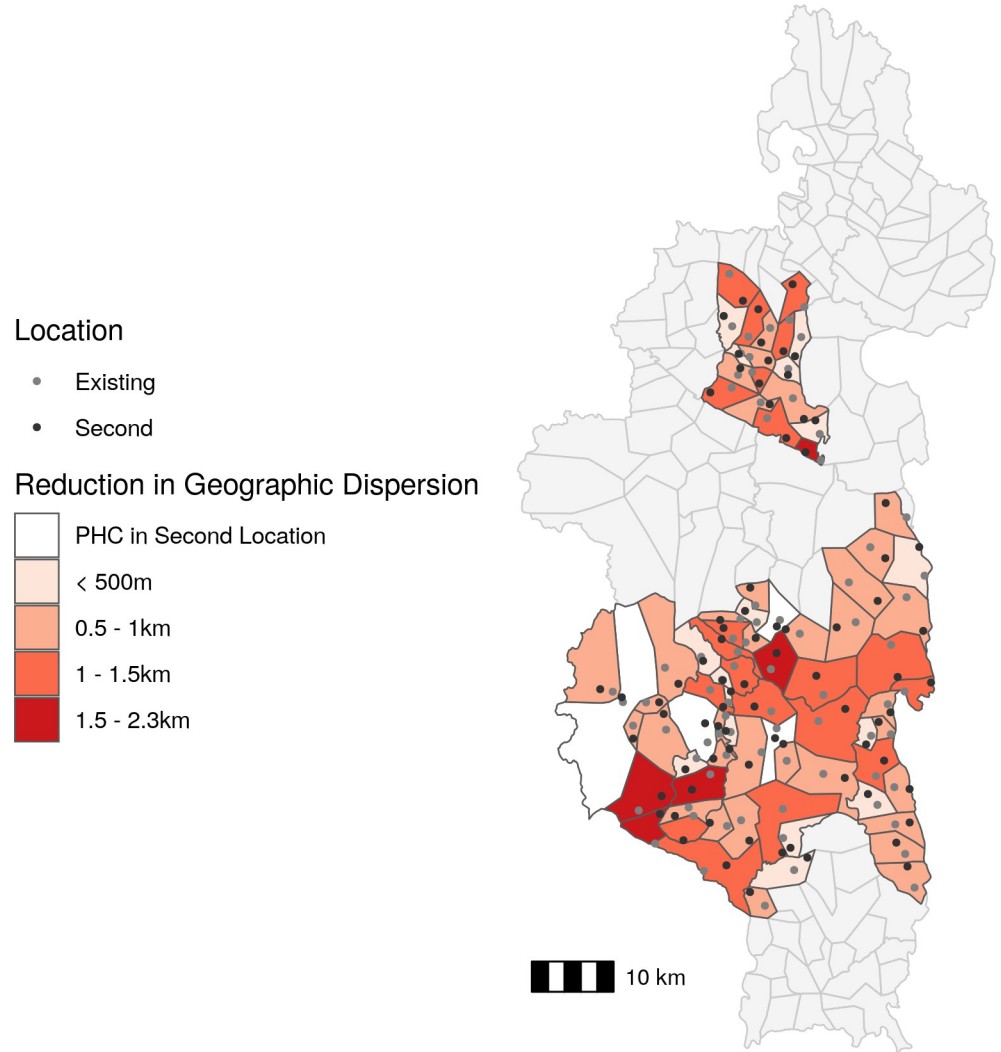

**Fig 5. The addition of a second community health site reduces geographic dispersion by over 500m for the majority of fokontany.** Absolute change in CHW geographic dispersion following addition of a second site in an optimal location compared to current geographic dispersion. The fill color represents the reduction in geographic dispersion and points represent existing site locations (gray) and second site locations (black). CHW: community health worker, PHC: primary health center. Data Sources: MMoPH (administrative boundaries), Pivot (health center locations), OpenStreetMap (transportation routes, buildings). All maps have been created from spatial datasets using the R programming language.

proactive care programs, may be needed to significantly reduce geographic barriers to community health access.

We found that geographic barriers to healthcare access persist in Ifanadiana, even at the level of community healthcare. Other studies in Ifanadiana have found that, despite the installation of an intervention that has virtually removed user fees and improved the quality of care at all levels of the health system, geographic barriers persist and negatively impact hospital referrals [49], use of primary health centers [9], child vaccination coverage [50], and malaria case ascertainment [51]. In addition, results from the IHOPE cohort in Ifanadiana showed that, while financial inequalities in the intervention catchment had been significantly reduced between 2014 and 2018, geographic inequalities persisted for most coverage indicators evaluated [29]. This trend is not unique to Madagascar, and negative impacts of geographic

accessibility on primary and secondary healthcare access are well studied (see Guagliardo [52] for a review). The findings of this study go one step further and show that geographic isolation remains a barrier for CHW access, where distances to seek care are much smaller. This contributes to a nascent, but growing literature studying the geography of community healthcare [53, 54], with the general consensus that geographic barriers also exist at this much finer spatial scale. Often, a threshold of 5km distance or 1 hour travel time is used to define spatial coverage of healthcare access (e.g. [4]), but we find that, for community care, a meaningful effect of distance on healthcare utilization remains even within that definition of "coverage".

Our results have important implications for the design of national community health programs in Madagascar and elsewhere. The community health policy of Madagascar in recent years required two CHWs per fokontany regardless of their size or geographic dispersion of the population. In late 2022, national community health guidelines were updated to recommend one CHW for every 300 households in a fokontany [36], although this policy has not yet been implemented. However, the guidelines do not include recommendations for the distribution of CHWs or health sites with respect to spatial patterns in population densities or transport networks. Our results suggest that in addition to the recommended ratios, the location of CHWs should be adapted to the local geography of CHW catchments, and additional interventions may be necessary to reach isolated communities. Our results also support the need for proactive community case management programs (pro-CCM), which have been implemented elsewhere [55], and were recently recommended in the MMoPH's community health policy. Under pro-CCM, CHWs visit each household within a catchment at regular intervals to provide basic services and conduct educational campaigns in addition to other elements, such as cost-free care, CHW professionalization and case follow-up regimes [56, 57]. Given the strong association between high geographical dispersion and low consultation rates, the implementation of pro-CCM programs in Madagascar has great potential for improving health outcomes.

Geographic optimization has been proposed as one potential solution to reduce barriers to healthcare [33, 53], but our optimization exercise predicts that optimizing the location of single community health sites would only marginally improve consultation rates. There are several potential explanations for this. First, the majority of fokontany already had CHW sites in geographically optimal locations. Pivot staff reports that CHW site locations in Ifanadiana are usually chosen by the fokontany mayor, and are often in one of the larger villages in the fokontany, if not the largest. Because of this, a significant proportion of the population by default lives within the same residential area as the CHW site, resulting in lower geographic dispersion. This implies that communities are able to identify optimal locations without the use of a complex geographic algorithm, and should be included in decision making regarding community health programs. Second, in Ifanadiana, communities in larger fokontany are necessarily further from community health sites (Fig 1). Given the larger area of these fokontany, there is a limit to how much optimizing placement can reduce geographic dispersion. Installing a second community health site in a geographically optimized location, however, resulted in much higher predicted increases in consultation rates than optimizing one site alone. This effect was greatest in the most dispersed fokontany, and such a solution could be targeted to large, dispersed fokontany where one site cannot adequately serve the whole population [58]. Another alternative solution is the creation of pro-CCMs, where CHWs visit all households in their catchment on a regular basis [32, 55]. Proactive community care programs have been associated with reductions in child mortality rates and increases in access to primary healthcare [55]. A pilot proactive community care program is currently being implemented in one commune in Ifanadiana district, with positive initial impacts on consultation rates and quality of care [32]. While the limited geographic scale of the program precludes formal statistical analysis,

preliminary results show that this program is qualitatively increasing consultation rates in the more dispersed fokontany (S3 Fig).

In addition to geographic variables, we included several confounding variables describing characteristics of the population and CHWs in our model that warrant discussion. We found that CHW consultation rates were lowest for wealthier populations, counter to the general findings that wealthy populations have higher access to care [59]. Indeed, evidence for this positive relationship exists in Ifanadiana at the level of PHCs [29]. However, because we focused only on consultation rates with CHWs, we captured healthcare access for lower income populations that may not use PHCs because of financial barriers, and therefore rely more on CHWs for primary care. This suggests that CHWs are meeting their goal of serving vulnerable populations missed by other levels of the health system. In addition, we found that the gender make-up of CHWs influenced consultation rates, with consultation rates highest for all-men or mixed-gender teams. Historically, the majority of CHWs in Ifanadiana have been men, although there has been a focus on increasing gender equality among CHWs following international guidance from the WHO [60]. Following this guidance, there has been some concern that established gender norms in a field previously dominated by men could limit the effectiveness of women CHWs [61–63]. Our finding that mixed gender teams have the highest consultation rates suggests that including CHWs of multiple genders at each site is a potential way to increase gender equality without negatively impacting healthcare seeking behaviors, as has been suggested elsewhere [61]. Additional study, particularly using qualitative methods, is needed to further explore patient perceptions of CHW gender in Ifanadiana.

Here, we advance new methods to study the impact of geographic barriers at the community health level, a topic that is currently under-researched due to intrinsic challenges in the spatial resolution of data at these finer scales. Our use of a comprehensive geographic dataset with all the transportation routes and buildings in the district, publicly available on OpenStreetMap, allowed us to examine spatial patterns of healthcare access within CHW catchments with remarkable accuracy. Ihantamalala et al. [33] found that these methods more accurately represented residents' travel time than cost-distance maps often used to estimate potential access [4]. Recent studies of healthcare access [53, 64] have made use of fine-scale geographic data from OpenStreetMap or mobile phone data to obtain more accurate estimations of populations' geographical isolation. Several of these data sources are open-source and available online. For example, OpenStreetMap is an open-source dataset of transportation networks that can be viewed via a website (https://www.openstreetmap.org/) or downloaded via the Overpass API (https://overpass-turbo.eu/). In addition, Maina et al. [65] collated the location of over 98,000 public health facilities from 50 sub-Saharan African countries, which is available through the WHO's Global Malaria Programme (https://www.who.int/malaria/areas/surveillance/public-sector-health-facilities-ss-africa/en/). At the primary healthcare level, Weiss et al. [4] estimated travel times to these health facilities. Given the increasing availability of these data, future studies should leverage this fine-scale data to examine spatial patterns in healthcare inequalities, identifying underserved communities that may be otherwise missed in analyses that aggregate spatial data to coarser scales [66].

Our study had several limitations. CHW consultation data were reported at the level of the fokontany and individual visits are not georeferenced to each patient's residence. Therefore, our conclusions are limited to inference at the scale of the fokontany, not the individual. Collecting patient-level data would allow for the consideration of more detailed spatial patterns of healthcare utilization within the fokontany, such as identifying a distance threshold for catchment populations where CHW utilization is lower than average. Additionally, we infer the wealth of 80 fokontany from surveys in a sample of 1600 households that was not necessarily representative of those fokontany. Nevertheless, the distribution of wealth in the eighty spatial

clusters of the IHOPE cohort had a very consistent and strong spatial structure (e.g. higher wealth in areas near the paved road and around the larger towns) that justifies our spatial interpolation methods. Finally, we only examined one barrier to healthcare: geographic distance. There are many other factors that influence care-seeking behavior and access to care, including patient perceptions of care and treatment [67], costs of care [59], and CHW gender [68]. We partially addressed these other factors in our model by including variables for CHW age and gender and in the supplemental analysis that includes a measure of quality of care. However, the objective of this analysis was an in-depth consideration of geographic barriers, and future studies should consider these other factors in equal detail.

In conclusion, our results support WHO's recent guidelines for community health programs, which suggest that the local geography of CHW catchments be taken into account in program design and implementation. We show that geographic barriers to care can persist at the community health level and we advance new methods to better understand these barriers in local settings. More research on geographic access to healthcare at multiple levels of the health system, particularly at finer spatial-scales, is needed to progress towards the collective goal of reaching "everyone, everywhere" included in international UHC policies.

## Supporting information

**S1 Fig. Wealth scores are stable over time.** Correlation plots of wealth scores for the 80 clusters in the longitudinal cohort across the three sample years. Wealth scores are the mean of households in that cluster.
(TIF)

**S2 Fig. Results from the primary model are robust to the inclusion of a metric of quality of care.** Comparison of standardized log-coefficients from the primary model and the supplemental 'quality of care' model on a subset of the data. Points represent mean coefficient and error bars 95% CIs.
(TIF)

**S3 Fig. Proactive CHW programs increased consultation rates across all levels of geographic dispersion.** Points represent the percent change in consultation rates for each month of the year by comparing pre-proactive care (2017 and 2018) and post-proactive care (2020) for the eight fokontany in Ranomafana commune. Red diamonds represent the median per fokontany and shaded bars represent the 95% CI.
(TIF)

**S1 Table. Distribution of fokontany sizes in Ifanadiana district.**
(DOCX)

**S2 Table. Summary of HSS intervention carried out by PIVOT in Ifanadiana district, based upon TIDieR guidance.**
(DOCX)

**S1 File. Reflexivity statement.**
(DOCX)

**S1 Data.**
(DOCX)

## Acknowledgments

We would like to thank Pivot staff whose work in Ifanadiana provides the infrastructure that makes this research possible. We are thankful for the contributions of Vincent Herbreteau, Christophe Revilion, Jeremy Commin and the team of local mappers that contributed to the cartography work on OpenStreetMap in Ifanadiana. We would also like to thank the communities of Ifanadiana for their participation in the IHOPE surveys and the community health workers and supervisors for their efforts contributing to data collection in addition to providing primary health care to their communities.

## Author Contributions

**Conceptualization:** Michelle V. Evans, Tanjona Andréambeloson, Laura Cordier, Karen Finnegan, Feno Hanitriniaina, Andry Randriamahasoa, Bénédicte Razafinjato, Isaïe Jules Andriamiandra, Matthew H. Bonds, Andres Garchitorena.

**Data curation:** Michelle V. Evans, Tanjona Andréambeloson, Mauricianot Randriamihaja, Felana Ihantamalala, Laura Cordier, Ann C. Miller, Lanto Marovavy Ralantomalala, Bénédicte Razafinjato, Emeline Razanahanitriniaina, Rado J. L. Rakotonanahary, Andres Garchitorena.

**Formal analysis:** Michelle V. Evans, Tanjona Andréambeloson, Mauricianot Randriamihaja, Felana Ihantamalala, Andres Garchitorena.

**Funding acquisition:** Andres Garchitorena.

**Investigation:** Michelle V. Evans, Tanjona Andréambeloson, Laura Cordier, Karen Finnegan, Feno Hanitriniaina, Andry Randriamahasoa, Rado J. L. Rakotonanahary, Matthew H. Bonds.

**Methodology:** Michelle V. Evans, Tanjona Andréambeloson, Laura Cordier, Karen Finnegan, Feno Hanitriniaina, Andry Randriamahasoa, Bénédicte Razafinjato, Rado J. L. Rakotonanahary, Andres Garchitorena.

**Project administration:** Andres Garchitorena.

**Supervision:** Laura Cordier, Giovanna Cowley, Karen Finnegan, Ann C. Miller, Lanto Marovavy Ralantomalala, Emeline Razanahanitriniaina, Rado J. L. Rakotonanahary, Isaïe Jules Andriamiandra, Matthew H. Bonds, Andres Garchitorena.

**Validation:** Michelle V. Evans, Feno Hanitriniaina, Andry Randriamahasoa, Bénédicte Razafinjato, Rado J. L. Rakotonanahary.

**Visualization:** Michelle V. Evans.

**Writing – original draft:** Michelle V. Evans, Andres Garchitorena.

**Writing – review & editing:** Michelle V. Evans, Tanjona Andréambeloson, Mauricianot Randriamihaja, Felana Ihantamalala, Laura Cordier, Giovanna Cowley, Karen Finnegan, Feno Hanitriniaina, Ann C. Miller, Lanto Marovavy Ralantomalala, Andry Randriamahasoa, Bénédicte Razafinjato, Emeline Razanahanitriniaina, Rado J. L. Rakotonanahary, Isaïe Jules Andriamiandra, Matthew H. Bonds, Andres Garchitorena.

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
