## [Decision Letter · Decision Letter 0]

20 Oct 2022

PGPH-D-22-01308

Geographic barriers to care persist at the community healthcare level: evidence from rural Madagascar

Dear Dr. Evans,

Thank you for submitting your manuscript to PLOS Global Public Health. After careful consideration, we feel that it has merit but does not fully meet PLOS Global Public Health’s publication criteria as it currently stands. Therefore, we invite you to submit a revised version of the manuscript that addresses the points raised during the review process.

We look forward to receiving your revised manuscript.

Kind regards,

Hassan Haghparast Bidgoli

Academic Editor

Journal Requirements:

1. Please send a completed 'Competing Interests' statement, including any COIs declared by your co-authors. If you have no competing interests to declare, please state "The authors have declared that no competing interests exist". Otherwise please declare all competing interests beginning with the statement "I have read the journal's policy and the authors of this manuscript have the following competing interests:"

2. Please provide your detailed Financial Disclosure statement. This is published with the article. It must therefore be completed in full sentences and contain the exact wording you wish to be published.

a. Please clarify all sources of funding (financial or material support) for your study. List the grants (with grant number) or organizations (with url) that supported your study, including funding received from your institution. 

b. State the initials, alongside each funding source, of each author to receive each grant.

c. State what role the funders took in the study. If the funders had no role in your study, please state: “The funders had no role in study design, data collection and analysis, decision to publish, or preparation of the manuscript.”

d. If any authors received a salary from any of your funders, please state which authors and which funders.

3. Please provide separate figure files in .tif or .eps format.

4. We have noticed that you have uploaded Supporting Information files, but you have not included a list of legends. Please add a full list of legends for your Supporting Information files after the references list. 

Additional Editor Comments:

- Please provide more details on the context of the study, including some general SES of the setting, population health indicators in particular children, and health delivery system.

- Please also provide a summary of evidence, if available, on the main barriers to access to PHC in the setting and justify why you have focused on geographic distance in your study.

Reviewers' comments:

Reviewer's Responses to Questions

**Comments to the Author**

1. Does this manuscript meet PLOS Global Public Health’s publication criteria? Is the manuscript technically sound, and do the data support the conclusions? The manuscript must describe methodologically and ethically rigorous research with conclusions that are appropriately drawn based on the data presented.

Reviewer #1: Yes

Reviewer #2: Yes

2. Has the statistical analysis been performed appropriately and rigorously?

Reviewer #1: Yes

Reviewer #2: I don't know

3. Have the authors made all data underlying the findings in their manuscript fully available (please refer to the Data Availability Statement at the start of the manuscript PDF file)?

Reviewer #1: Yes

Reviewer #2: Yes

4. Is the manuscript presented in an intelligible fashion and written in standard English?

Reviewer #1: Yes

Reviewer #2: Yes

5. Review Comments to the Author

Reviewer #1: This is an interesting study of public health importance, using an innovative approach to investigating access to community-based healthcare.

Line 92: As a stand-alone paper, a brief description of the health system in Madagascar would be helpful for readers who are not familiar with that country, to provide context. One should not have to look up previous publications from this project for this information. What is the total population of Madagascar and percentage of rural vs urban? What constitutes primary care in this region? How is the health system funded? Are patients required to pay for care and at what level? (see line 466 and reference to financial barriers to PHC, line 462 do wealthier people use private services rather than community services?). How does the referral system from CHW to PHC work? Also a sense of topography would be helpful – is Madagascar a hilly country, are there physical obstructions between rural services such as large rivers? What kind of non-motorised transport do people use to access services?

Line 502: What are the main factors affecting access to PHC in public sector facilities of Madagascar? It would be helpful to have these briefly described in the introduction and some justification for the focus on geographic distance. It is not sufficient to give references to other publications for this context.

Line 97: it would be useful to see what the WHO recommended standards for workforce/population ratios and distances to the nearest health facility are. Does the term “medical professional” mean a doctor? If not, then the term health professional would be less confusing.

Line 99: it would be helpful to see what the child health indicators for Madagascar are, to get a sense of PHC use and context, for instance immunization coverage at age 5 years, infant and child mortality, chronic malnutrition ie stunting, and rates of childbirth at health facilities with skilled birth attendants (indicator of PHC coverage and neonatal health).

Line 112: “strong distance-decay effect on healthcare access” – how is this measured? These concepts may benefit from more explanation for readers who are not so familiar with this technique.

Line 180: CHW monthly reports - how reliable is this data? What checks were made to ensure that these data were trustworthy?

Line 192: It is worth repeating the time period of the study as a reminder to the reader.

Line 246: Is the tool available only in French? Is this the language used locally by CHW supervisors?

Line 259: the reflexivity statement is well written, a good model and will be appreciated by readers.

Line 264: Is there any information on what the nature of these consultations were? Were all consultations captured or only those with children? What was the range? If the average annual consultation rate was 1.73 consultations per capita for children only, this may be low. What was expected? Is there anything to compare with?

Line 292: “a decrease in annual consultation rates from 2.18 to 0.81 per capita” – where did 2.18 come from? The annual consultation rate per capita was given as 1.73 earlier.

Line 488: “the increasing availability of these data” - how available is this data to most countries in Africa? Readers would be interested to know how to access systems that generate such data. It would be useful to be able to map a range of influential PHC factors such as access to safe water and sanitation facilities, roads, schools, local government offices, as well as clinics and personnel which also contribute to “spatial patterns in healthcare inequalities, identifying underserved communities” (line 489).

In conclusion, the paper needs to address more robustly the “so what?” question, what is the lesson for the PHC community? It seems obvious that more CHWs will be better than less, especially in dispersed areas of low population density (line 101:Two volunteer CHWs per fokontany, population of approx. 1200, but with a range in size from 1.5 to 3747 km2), the solution of distributing CHWs based on geographic equity rather than population numbers is well documented in the literature and apparently obvious to communities (see line 443: “communities are able to identify optimal locations without the use of a complex geographic algorithm). It is difficult to get a sense from this paper of how well-served Madagascar is in PHC resources, including numbers of CHWs, than other countries in Africa. Countries with well-developed CHW programs usually involve CHWs who travel to households rather than being in static sites, as a principle of removing barriers of transport etc, but then need transport and other resources. They are also then more familiar with conditions that their patients come from. It would be interesting to unpack the gender dynamics more – do the communities think male CHWs are doctors (a phenomenon in many countries) and therefore consult them more? Do women feel satisfied consulting male CHW for pregnancy related care? This may be an area for further research. To persuade readers of the usefulness of this technique of measuring geographic access to PHC, rather than confirming the obvious, the authors could present a more detailed picture of PHC in Madagascar into which this new knowledge creates a better understanding of improving access to care (an indicator of quality of care).

Reviewer #2: General comment:

This is a valuable and important paper on a topic of great importance that has received little attention in the peer-reviewed literature – namely the optimal geographic location of CHW and geographic influences on CHW utilization. It is simplistically assumed that geographic barriers to access care are removed when services are provided by CHWs, and this paper makes the important point that this is not necessarily the care, The authors are to be congratulated on this useful piece of research. The paper is well-written and well-organized.

Specific comments:

1. The analysis employs a lot of sophisticated geographic analyses that I am not qualified to comment on. On the surface, it looks like it has been carefully done.

2. It seems to me that the authors should raise the possibility of having the government assign more than 2 CHWs in those fokotanys that are larger in size and that have larger populations. This should be mentioned in the Discussion section, providing more information on the size and population variability among fokotanys would be useful for the Results section. Why not propose a more optimal model that doesn’t limit each fokotonay to just 2 CHW sites. Also, it would be interesting to add a table that lists the number of fokotanys by category or size and population.

3. Likewise, the Discussion section should also raise the possibility of relying more on home visits from CHWs rather than simply expecting the beneficiaries to come to the CHW sites. The authors briefly mention in the Discussion section the power of Pro-CCM, but the advantages of home visitation more broadly could be emphasized and strengthened beginning on line 414. It is good to see that you are testing this out in a pilot area! Here are some references that support that:

Johnson, A. D., Thiero, O., Whidden, C., Poudiougou, B., Diakite, D., Traore, F., . . . Kayentao, K. (2018). Proactive community case management and child survival in periurban Mali. BMJ Glob Health, 3(2), e000634. doi:10.1136/bmjgh-2017-000634

Perry, H. B., Rassekh, B. M., Gupta, S., & Freeman, P. A. (2017). Comprehensive review of the evidence regarding the effectiveness of community-based primary health care in improving maternal, neonatal and child health: 7. shared characteristics of projects with evidence of long-term mortality impact. J Glob Health, 7(1), 010907. doi:10.7189/jogh.07.010907

4. On line 67, another appropriate reference to cite there is:

Perry, H. (2020). Health for the People: National Community Health Programs from Afghanistan to Zimbabwe. Retrieved from https://pdf.usaid.gov/pdf_docs/PA00WKKN.pdf and https://chwcentral.org/wp-content/uploads/2021/11/Health_for_the_People_Natl_Case%20Studies_Oct2021.pdf

5. On line 172, how were the clusters selected?

6. PLOS authors have the option to publish the peer review history of their article (what does this mean?). If published, this will include your full peer review and any attached files.

**Do you want your identity to be public for this peer review?** For information about this choice, including consent withdrawal, please see our Privacy Policy.

Reviewer #1: No

Reviewer #2: No

---

## [Decision Letter · Decision Letter 1]

28 Nov 2022

Geographic barriers to care persist at the community healthcare level: evidence from rural Madagascar

PGPH-D-22-01308R1

Dear Dr. Evans,

We are pleased to inform you that your manuscript 'Geographic barriers to care persist at the community healthcare level: evidence from rural Madagascar' has been provisionally accepted for publication in PLOS Global Public Health.

Best regards,

Hassan Haghparast Bidgoli

Academic Editor

Thanks for addressing all reviewers' comments. There is no additional comments from the reviewers and editor.

Reviewer Comments (if any, and for reference):

Reviewer's Responses to Questions

**Comments to the Author**

1. If the authors have adequately addressed your comments raised in a previous round of review and you feel that this manuscript is now acceptable for publication, you may indicate that here to bypass the “Comments to the Author” section, enter your conflict of interest statement in the “Confidential to Editor” section, and submit your "Accept" recommendation.

Reviewer #1: All comments have been addressed

Reviewer #2: All comments have been addressed

2. Does this manuscript meet PLOS Global Public Health’s publication criteria? Is the manuscript technically sound, and do the data support the conclusions? The manuscript must describe methodologically and ethically rigorous research with conclusions that are appropriately drawn based on the data presented.

Reviewer #1: (No Response)

Reviewer #2: Yes

3. Has the statistical analysis been performed appropriately and rigorously?

Reviewer #1: (No Response)

Reviewer #2: I don't know

4. Have the authors made all data underlying the findings in their manuscript fully available (please refer to the Data Availability Statement at the start of the manuscript PDF file)?

Reviewer #1: (No Response)

Reviewer #2: Yes

5. Is the manuscript presented in an intelligible fashion and written in standard English?

Reviewer #1: (No Response)

Reviewer #2: Yes

6. Review Comments to the Author

Reviewer #1: (No Response)

Reviewer #2: The authors have done a good job of responding to my comments in a way that strengthens the paper. In addition, the authors' response to the excellent review by Reviewer 1 has also greatly strengthened the paper. I recommend its acceptance.

7. PLOS authors have the option to publish the peer review history of their article (what does this mean?). If published, this will include your full peer review and any attached files.

**Do you want your identity to be public for this peer review?** For information about this choice, including consent withdrawal, please see our Privacy Policy.

Reviewer #1: No

Reviewer #2: **Yes: **Henry B. Perry
